# Blood Transcriptomic Analyses Reveal Functional Pathways Associated with Thermotolerance in Pregnant Ewes Exposed to Environmental Heat Stress

**DOI:** 10.3390/genes14081590

**Published:** 2023-08-06

**Authors:** Rosa I. Luna-Ramirez, Sean W. Limesand, Ravi Goyal, Alexander L. Pendleton, Gonzalo Rincón, Xi Zeng, Guillermo Luna-Nevárez, Javier R. Reyna-Granados, Pablo Luna-Nevárez

**Affiliations:** 1School of Animal and Comparative Biomedical Sciences, University of Arizona, Tucson, AZ 85721, USA; 2Genus R&D, Genus PLC, DeForest, WI 53532, USA; 3Zoetis Inc., VMRD Genetics R&D, Kalamazoo, MI 49007, USA; 4Departamento de Ciencias Agronómicas y Veterinarias, Instituto Tecnológico de Sonora, Ciudad Obregón 85000, Mexico

**Keywords:** genes, heat stress, sheep, thermotolerance, transcriptomic

## Abstract

Environmental heat stress triggers a series of compensatory mechanisms in sheep that are dependent on their genetic regulation of thermotolerance. Our objective was to identify genes and regulatory pathways associated with thermotolerance in ewes exposed to heat stress. We performed next-generation RNA sequencing on blood collected from 16 pregnant ewes, which were grouped as tolerant and non-tolerant to heat stress according to a physiological indicator. Additional samples were collected to measure complete blood count. A total of 358 differentially expressed genes were identified after applying selection criteria. Gene expression analysis detected 46 GO terms and 52 KEGG functional pathways. The top-three signaling pathways were p53, RIG-I-like receptor and FoxO, which suggested gene participation in biological processes such as apoptosis, cell signaling and immune response to external stressors. Network analysis revealed *ATM*, *ISG15*, *IRF7*, *MDM4*, *DHX58* and *TGFβR1* as over-expressed genes with high regulatory potential. A co-expression network involving the immune-related genes *ISG15*, *IRF7* and *DXH58* was detected in lymphocytes and monocytes, which was consistent with hematological findings. In conclusion, transcriptomic analysis revealed a non-viral immune mechanism involving apoptosis, which is induced by external stressors and appears to play an important role in the molecular regulation of heat stress tolerance in ewes.

## 1. Introduction

Climate change has increased global temperature, as well as the frequency of extreme environmental events, such as droughts, which intensify warm conditions, leading to heat stress in domestic livestock, impacting agricultural productivity and affecting food security [1]. In semiarid regions, livestock have to cope with heat stress and reduced water availability, while maintaining an acceptable level of production to ensure profitability. Therefore, their ability to tolerate warm environments has become an important selection criterion to promote fecundity of thermotolerant sheep, especially under semiarid climates characterized by environmental heat stress during most of the year [2].

Thermotolerance in sheep exposed to heat stress could be defined as the ability to cope, adapt or reduce their vulnerability to adverse effects of elevated temperatures, without affecting animal productive performance. A genetic component associated with such ability has been reported for dairy cattle, beef cattle and sheep [3,4]. Further studies are needed to decipher the genetic bases that regulate heat stress tolerance in sheep, which first requires us to identify adequate and easily measurable indicators of thermotolerance [5]. Several behavioral and physiological markers have been used to characterize thermotolerance in sheep. The markers most commonly tested are feed intake (FI), rectal temperature (RT) and respiration rate (RR), as well as endocrine markers such as cortisol, prolactin and thyroid hormones [6]. 

Recently, studies have shown a decline in production traits after an animal exceeds its thermoneutral threshold, and these findings indicate a close relationship between heat stress and animal performance [7]. Thus, in order to use a more reliable measurement of heat stress response, a thermotolerance indicator involving physiological traits (i.e., RT and RR) and milk yield was constructed to investigate genes and molecular mechanisms associated with heat stress in dairy cattle [8]. Similarly, RT and FI data collected from ewes exposed to heat stress were used to construct a thermotolerance indicator using a simple regression model [9]. A genomic analysis of the indicator based on RT and FI identified three significant SNPs that were further validated as markers for heat tolerance and fertility in an independent population of sheep [10].

Transcriptomic analyses in sheep have identified differentially expressed genes (DEGs) between ewes exposed and non-exposed to heat stress conditions [11,12,13]. Similarly, recent RNA-Seq studies have reported DEGs associated with breed-sensitivity to heat stress in sheep [14,15]. However, tolerance to heat stress exposure in sheep has not been studied thoroughly with transcriptomic technologies. 

RNA sequencing experiments in sheep exposed to heat stress have been performed using different tissues, which include hypothalamus [11], pituitary [12], liver [13,16] and muscle [17]. However, the blood transcriptome offers some advantages because of the accessibility and minimal invasiveness during sampling [18], as well making it easy to study and identify physiological markers [19]. In this context, blood is a potentially dynamic tissue that reflects the physiological status of the animal [20]. Moreover, blood transcriptomic markers have been used to characterize vascular diseases [21], immune system responses [22], obesity status [23] and aging effects [24]. 

Therefore, our objective was to identify DEGs that explain the molecular basis of thermotolerance in blood from pregnant ewes classified as heat tolerant (HT) and non-heat tolerant (NHT) after a one-week exposure to environmental heat stress. Furthermore, functional analyses of the DEGs helped to decipher the molecular mechanism regulating tolerance to heat stress in ewes.

## 2. Materials and Methods

The University of Arizona Institutional Animal Care and Use Committee (IACUC) approved all procedures performed in animals (Approval Code: 12-401; 14 September 2021). Animals were managed in compliance with the Guide for the Care and Use of Laboratory Animals (8th ed.).

### 2.1. Animals and Management

The current research included data and phenotypes from 16 time-mated, pregnant Columbia–Rambouillet crossbred ewes, with good body condition score, 2 to 4 years old and weighing 63 ± 3 kg. Singleton pregnancies were confirmed by ultrasonography at ~35 days of gestation age. Ewes were housed in a room where climate conditions were controlled and recorded by the Distech Building Automation system (Jenco Co., Tucson, AZ, USA). This software was programmed to provide thermo-neutral conditions during a 5-day acclimation period (23 °C and 35% RH; THI = 21). Then, all ewes were heat-challenged by exposure to an environmental temperature of 40 °C between 6 AM and 6 PM (RH = 36%, THI = 34.9) and 35 °C between 6 PM and 6 AM (RH = 47%, THI = 31.6), with the dew point set at 22 °C [25]. Ewes were subjected to heat stress starting at day 39 ± 1 of gestation, continuing through day 96 ± 1 (i.e., 57 days; term = 149 dGA). 

Standard-bread alfalfa pellets (Sacate Pellet Mills) were provided to all ewes with ad libitum access to feed, salt and water. Feed and water intake were recorded daily inside the climate chamber using an electronic scale and meter water buckets, respectively. Individual body weight was recorded prior to heat stress exposure using a walkover scale. Rectal temperature was collected daily at 10 AM using a digital thermometer. 

We used a heat-stressed pregnant ewe model because it represents a well-developed, reproducible laboratory model to study physiological thermotolerance response under controlled heat-stressed conditions. This natural model has been used to establish placental insufficiency and fetal growth restriction, and fetal and neonatal physiological results have been reported [26,27,28]. 

### 2.2. Climatic Data and Thermo-Tolerance Indicator

Ambient temperature (AT) and relative humidity (RH) data collected inside the chamber were used to calculate the temperature–humidity index (THI), which estimates the level of heat stress (HS) to which the ewes were exposed, using the formula THI = AT °C − [(0.31 − 0.31 RH %) (AT °C − 14.4)] [29]. The values obtained indicate the following: <22.2 = absence of HS; 22.2 to <23.3 = moderate HS; 23.3 to <25.6 = severe HS and ≥25.6 = extremely severe HS.

Phenotypic records of feed intake (FI) and rectal temperature (RT) were collected daily from each ewe located inside the climate chamber. These data were processed to calculate a thermotolerance indicator by using a simple linear regression analysis that predicted FI as a function of RT, as described by Luna-Nevarez et al. [9].

### 2.3. RNA Isolation and Sequencing

The Columbia–Rambouillet crossbred pregnant ewes (*n* = 16), with good body condition and 2 to 4 years old were then selected for transcriptomic analyses. These ewes were exposed for 57 days to controlled heat-stressed conditions inside a climate chamber and subsequently classified as HT (*n* = 8) and NHT (*n* = 8) according to a thermotolerance indicator. Blood samples were individually collected from the jugular vein on days −1, 2, 4, 7, 14 and 28 of heat stress exposure. These samples were frozen and later sent to Zoetis Inc. Lab for RNA extraction and next-generation sequencing. We identified the greatest response in differential gene expression on day 7. Therefore, only RNA-Seq data from day 7 were retained for subsequent analyses and presented in the current study.

Blood samples collected from heat-stressed ewes and previously frozen in PAXgene tubes were thawed at room temperature overnight. RNA was isolated on Qiagen’s QIAsymphony using Qiagen’s QIAsymphony PAXgene Blood RNA Kit (Qiagen Ltd., Germantown, MD, USA) following the manufacturer’s protocol. Concentration of RNA was determined using both Qubit (Life Technologies, Carlsbad, CA, USA) and Nanodrop (Thermo Scientific, Waltham, MA, USA), and quality was determined via Agilent’s 4200 TapeStation (Agilent Technologies, Santa Clara, CA, USA). Then, 1 μg of RNA was used as input into Illumina’s TruSeq Stranded mRNA Library Prep kit, in which analysis was performed according to the manufacturer’s protocol. Library construction quality was assessed via Agilent’s 4200 TapeStation and KAPA Library Quantification qPCR kit (Roche, Indianapolis, IN, USA). Libraries were normalized, pooled and run on Illumina’s NextSeq500 (Illumina, San Diego, CA, USA) using a high output 75 bp kit for a 1 × 75 bp run, with an aim of obtaining 25 million reads per sample.

Libraries were loaded onto NextSeq500 to sequence the template fragments in both the forward and reverse directions. On a flow cell, samples were bound to complementary adapter oligos using the kit reagents. The adapters were made such that, after the reverse strand was re-synthesized during sequencing, they could be used to selectively cleave the forward strands. The opposite end of the fragment was then sequenced using the copied reverse strands. This sequencing by synthesis technique allows for the detection of fluorescently labeled nucleotide bases, as they are incorporated into DNA template strands. Each sequencing cycle contains all 4 reversible terminator-bound dNTPs.

### 2.4. Raw Data Reads and Alignment

Raw data reads were obtained after RNA sequencing in FastQC format. FastQC software (version 0.11.9, https://www.bioinformatics.babraham.ac.uk/projects/fastqc/ (accessed on 14 November 2022)) was used to conduct quality control of sequence data. Resulting low quality reads, as well as reads containing standard adapters or poly-N sequences, were trimmed using Trimmomatic tool (version 0.36). Only clean reads that passed the quality control and averaged base quality higher than 20 were selected for subsequent analyses. The Spliced Transcripts Alignment to a Reference (STAR) software (version 2.5.3a) was used to align high-quality paired-end reads to the reference genome of sheep (*Ovis aries* Oar_Rambouillet_v1.0.102) downloaded from Ensembl website (http://ftp.ensembl.org/pub/release-102/fasta/ovis_aries_rambouillet/DNA/ (accessed on 6 December 2022)). 

### 2.5. Differential Gene Expression Analyses

Mapped reads in BAM format were annotated and counted using featureCounts (v1.6.0) along with the gencode comprehensive gene annotation (gencode Oar_rambouillet_v1.0.102 gtf file) [30]. Fragments per kilo base of transcript per million mapped fragments (FPKM) of each sample were counted to estimate gene expression levels [31]. The resulting text file containing a matrix with the number of transcribed genes per sample was used as input for running the DE analysis. Differential gene expression testing was performed using the package DESeq2 within Bioconductor (https://bioconductor.org/packages/release/bioc/html/DESeq2.html (accessed on 21 January 2023)) in R Version 4.1.

Genes with very low counts per million (CPM ≤ 1) were filtered out. DEGs between the compared groups, i.e., heat stress tolerant (HT) and non-heat stress tolerant (NHT), were analyzed with the general linear models (GLM) included in DESeq2. Data were normalized internally using DESeq2’s median of ratios method. The Benjamini–Hochberg procedure for controlling the false discovery rate (FDR) was used to adjust *p*-values [32]. Therefore, genes differentially expressed between HT and NHT groups were defined by a threshold of adjusted *p*–value < 0.05, Log2FC > 1 or Log2FC < −1 and BaseMean > 2. 

### 2.6. Network and Functional Analysis

The Search Tool for the Retrieval of Interacting Genes/Proteins (STRING) database (https://string-db.org/ (accessed on 8 February 2023)) was used to perform a gene network analysis through the protein–protein interactions in a list of the DEGs [33]. A centrality analysis was performed from the gene network created by the STRING database, in order to identify the genes with a higher number of connections in the network. The IMEx Interactome tool (high confident score) from NetworkAnalyst software v3.0 (https://www.networkanalyst.ca/ (accessed on 15 February 2023)) was used to confirm the genes with the highest regulatory potential, which was based on their interactions with other proteins [34,35]. This software was also used to construct gene co-expression networks within each of the different blood cellular components, in order to study the gene’s functionality at the level of the blood tissue.

The Database for Annotation, Visualization and Integrative Discovery (DAVID, https://david.ncifcrf.gov/ (accessed on 19 February 2023)) was used to perform the Gene Ontology (GO) enrichment analysis and to detect the biological functions of the DEGs. Genes resulting from the GO analysis were grouped into the three main GO categories: cellular components, molecular functions and biological processes. *p*-values were calculated using a hypergeometric distribution before multiple testing and Benjamini–Hochberg correction. Kyoto Encyclopedia of Genes and Genomes (KEGG, http://www.genome.jp/kegg/ (accessed on 1 March 2023)) pathway analyses were also performed using the DAVID database to detect significant pathways associated with DEGs [36]. A threshold of FDR < 0.05 was used to obtain highly significant metabolic pathways.

### 2.7. Validation of RNA-Seq Data Using Quantitative RT-PCR

RNA sequencing findings were validated with quantitative polymerase chain reaction (qPCR). Primer blast was used to design oligonucleotide primers from the sheep genome. Primers’ optimal annealing temperatures and primer efficiencies were determined using PCR. Primer specificity was confirmed with nucleotide sequencing of the clones’ PCR product (PCR 2.1-TOPO vector, Thermo Fisher Scientific Life Sciences, Waltham, MA, USA). cDNA was prepared by using 0.4 µg total RNA (per reaction) using Superscript III (Thermo Fisher). PCR products were amplified using SYBR Green (Qiagen, Valencia, CA, USA) in an iQ5 Real-Time PCR Detection System (Bio-Rad Laboratories). All qPCR results were normalized to the geometric mean of ribosomal protein s15 reference gene. Quantitative analysis was performed using the comparative Ct method (Ct gene of interest–Ct geometric mean of the reference gene), and fold change was calculated using the 2-ΔΔCT method [37].

### 2.8. Hematological Parameters 

Blood samples (2 mL) were drawn via venipuncture from another set of HT (*n* = 9) and NHT (*n* = 8) pregnant ewes on day 7 after being exposed to heat stress. Samples were collected into complete blood count/ethylenediaminetetraacetic (EDTA) acid tubes and sent to the Arizona Veterinary Diagnostic Laboratory (AVDL). The anticoagulated blood was used for manual complete blood counts (CBC) and differential white blood cells. Hematological parameters that were measured included white blood cells (WBC), red blood cells (RBC), hemoglobin (HGB), hematocrit (HCT), mean corpuscle volume (MCV), mean corpuscular hemoglobin (MCH) and mean corpuscular hemoglobin concentration (MCHC). The differential WBC included neutrophils, lymphocytes, monocytes, eosinophils and basophils count. Student’s *t*-test was performed to compare hematological values between HT and NHT ewes using *p* < 0.05 as the level of significance.

## 3. Results

### 3.1. Thermotolerance Indicator

The regression analyses performed on a subset of heat-stressed ewes (*n* = 16) revealed that rectal temperature resulted as predictor for feed intake (*p* < 0.05). The regression coefficient (β1) from the linear regression model was proposed as thermotolerance indicator, and it served to classify ewes as heat stress tolerant (HT) and non-heat stress tolerant (NHT). As observed in Table 1, HT ewes were able to increase feed intake (+β1) even though their rectal temperature showed a constant increase during heat stress exposure. Conversely, ewes that decreased their feed intake (–β1) as rectal temperature increased during exposure to heat stress were grouped as NHT.

### 3.2. Blood Cell Parameters

The average values for blood cell counts between HT and NHT ewes’ groups are presented in Table 2. White blood cell (WBC) count was higher (*p* < 0.05) in NHT compared with HT ewes. Similarly, ewes from the NHT group showed higher values for lymphocyte (*p* < 0.01) and monocyte (*p* < 0.05) cell counts than HT ewes. From red blood cell indices, only RBC count differed (*p* < 0.05) between HT and NHT ewes’ groups.

### 3.3. Read Mapping and Gene Expression

A total of 325,577,849 raw reads were generated from 16 samples. After quality control 274,136,549 clean reads were obtained (84.2%). The GC content was in accordance with base composition rules, as this value ranged from 48 to 52%. From the clean reads, approximately 62.67% had single aligned positions (i.e., uniquely mapped), and they were aligned to the sheep reference genome and retained for further bioinformatics analysis.

RNA-Seq analysis detected 18,883 transcripts expressed in blood samples collected from ewes exposed to heat stress. Only 358 genes fulfilled criteria established for DEGs between HT and NHT ewes’ groups; from these genes, 242 were upregulated and 116 were downregulated (Figure 1).

### 3.4. GO Enrichment and Functional Analyses

The three main GO categories (i.e., biological processes, cellular components and molecular function) were analyzed according to the list of DEGs between HT and NHT ewes. In total, we found 46 GO terms with *p* < 0.05, and the most significantly enriched terms appeared to be associated with cell signaling, apoptosis and immune response to external stressors (Table 3). Regarding biological processes, the most represented GO terms were regulation of TORC1 signaling, regulation of cell communication, type I interferon production, regulation of signal transduction, regulation of immune system processes, lipoprotein metabolic process, regulation of cytokine production and cellular metabolic processes. The main terms within cellular component GO group were bicellular tight junction and external side of plasma membrane, whereas the most represented GO terms within molecular function category were ATP binding and cytokine binding.

Differentially expressed genes between HT and NHT ewes showed enrichment (*p* < 0.05) of 52 pathways according to the KEGG pathway analysis. The most significantly enriched pathways that accounted for FDR correction (i.e., *p* < 0.05 after Benjamini–Hochberg adjustment) were p53-signaling pathway, RIG-I-like receptor signaling pathway and FoxO signaling pathway (Figure 2, Table 4). These pathways appear to be related to heat stress through activation of physiological functions such as cell signaling, cellular processes and immune response to external stressors. The eight other pathways were significant at *p* < 0.05 but did not accomplish FDR correction.

### 3.5. Network Analyses for DEGs

From the 358 DEGs, only 38 were grouped in the main network that was constructed using the STRING database (Figure 3) because they showed at least one connection. The centrality analysis performed with this network identified the genes *ATM*, *SMC2, ISG15*, *IRF7*, *MDM4*, *CDK14, TBL1RX1*, *LIN7C, DHX58, TGFβR1, RIF1, KIFB5, DYNCIII* and *ABCA1* as those showing larger connectivity and higher interactions with other genes (Table 5). The regulatory ability of these genes is visualized in Figure 4, which was obtained from IMEx Interactome. Interestingly, six of these genes belong to the enriched pathways detected in this study, which were p53-signaling pathway (*ATM*, *MDM4*), RIG-I-like receptor signaling pathway (*ISG15*, *IRF7*, *DHX58*) and FoxO signaling pathway (*TGFβR1*, *ATM*). Finally, blood cell-specific co-expression networks involving the immune-related genes *ISG15*, *IRF7* and *DXH58* were observed within lymphocytes and monocytes in WBCs (Figure 5A,B).

### 3.6. RNA-Seq Validation

Four DEGs were selected to validate the sequencing results of RNA-Seq (Figure 6) using a confirmatory analysis by qRT-PCR. For these four genes, the RNA-Seq expression profiles showed an identical expression pattern than qRT-PCR results. Moreover, the correlation analysis detected a positive and significant association (r = 0.95; *p* < 0.05) between qRT-PCR fold changes and RNA-Seq fold changes for the four genes. These results validated that the gene expression data estimated in the current study by RNA-Seq analysis were accurate and reliable.

## 4. Discussion

Sheep production in semiarid regions is challenging due to the warm temperatures that prevail most of the year. Once the ambient temperature increases above body temperature (37 °C), ewes are unable to dissipate internal heat, which leads to heat stress. Novel “omics” technologies have been proposed as a strategy to decipher genetic basis associated with heat tolerance and heat stress response.

Previous studies performed transcriptomic analyses in ewes, comparing heat stress and thermoneutral control groups in order to identify DEGs associated with heat stress. The main pathways that were enriched in these studies were P53, PI3K-Akt and PPAR signaling pathways, as well as metabolic pathways [11,12,13]. A transcriptomic study for heat stress response involving sheep breeds with different heat sensitivity detected enriched pathways mainly involved in signal transduction and energy metabolism [15]. Similarly, transcriptome comparison between indigenous and exotic sheep breeds revealed that their adaptability was associated with their immune response, which was mediated by difference secretory patterns of IgG and cytokines [14]. 

To our knowledge, this is the first transcriptomic study to compare heat-stressed ewes classified as heat tolerant and non-heat tolerant and report DEGs and their pathways. The indicator of heat stress tolerance used in the current study was previously reported by our research group because of its association with three candidate SNPs in a genome-wide association study [9]. These SNPs were further validated as markers for fertility and thermotolerance in an independent sheep population [10]. 

After day 7 of environmental heat stress, complete blood cell counts showed lower (*p* < 0.05) RBC numbers and greater (*p* < 0.05) WBC numbers in NHT ewes compared to HT ewes. The elevation in WBCs was mainly observed in lymphocytes and monocytes (*p* < 0.05). However, these changes in blood components did not exceed the normal ranges, which indicates the ewes were healthy.

Chronic heat stress increased the proportion of monocytes and B-lymphocytes in the spleen of pigs, which supports an enhanced immune response [38]. Monocytes are particularly responsive to heat stress compared to other WBC subtypes because a protein activated by heat stress, the heat-shock protein-72 (HSP72), increases monocyte expression as part of the inflammatory response following physiological stress [39]. Moreover, in chickens, heat stress increased serum corticosterone concentrations, which stimulated lymphocyte proliferation. This appears to be part of an adaptive and evolutionary response to improve immune surveillance after exposure to heat stress conditions [40].

Analyses of gene expression identified functional pathways associated with apoptosis, cell signaling and immune response to external stressors. Gene networks from DEGs detected co-expression of immune-related genes within lymphocytes and monocytes WBC. These findings are consistent with greater WBC counts and confirm a molecular immune mechanism in blood from ewes tolerant to heat stress. The top three signaling pathways identified in this study were p53-signaling pathway, RIG-I-like signaling pathway and FoxO signaling pathway. These pathways, as well as their related candidate genes, may have critical roles in the response to heat stress in HT ewes. Furthermore, this analysis begins to explain the mechanisms by which some ewes have the ability to tolerate heat stress but still retain normal productive and functional traits.

### 4.1. Apoptosis and Cell Signaling

Heat stress is a major extracellular stimulus resulting when ambient temperatures exceed body temperature in animals, making them unable to dissipate heat. The intense environmental heat increases cellular temperature and toxicity, leading to apoptosis and cell death, as a consequence of a protein denaturation that interrupts critical cellular processes [41]. The p53-signaling pathway is a key regulator of apoptosis, which, under heat-stressed conditions, works as a transcription factor to activate several genes related to the heat stress response [42]. In addition, oxidative stress and generation of reactive oxygen species (ROS) activate the pro-apoptotic function of the p53-signaling pathway [43]. 

In the current study, the p53-signaling pathway was enriched (according to KEGG pathway analysis). The genes that were associated with this pathway were *ATM, SESN3* and *MDM4,* which were upregulated in the HT group and appeared to be associated with tolerance to heat stress. 

Life is threatened due to heatstroke when the body temperature increases above 40 °C; under such conditions, p53 protein is phosphorylated by the gene *ATM* (ATM Serine/Threonine Kinase), leading to an increased expression of the gene *BAX* (BCL2 Associated X, Apoptosis Regulator) that induces apoptosis [44]. Hyperthermia induced by heat stress causes DNA damage, which suggests DNA repair is an essential process to maintain genomic integrity [45,46]. After exposure to heat stress, the master kinase ATM phosphorylates the histone H2AX and activates the appropriate cell cycle factors to repair chromatin domains surrounding the damaged DNA. The mechanism regulating this DNA repair involves the p53 signaling pathway. ATM is important because it initiates and coordinates the cellular response to DNA damage, and this protein also induces the recruitment of additional effector repair proteins [47].

Sestrins (i.e., SESN1, SESN2 and SESN3) are stress-inducible metabolic proteins that protect organisms against DNA damage, oxidative stress, starvation and hypoxia and play a pivotal role in apoptosis inhibition, a critical process involved in the response to heat stress [48]. Sestrins control important cellular functions associated with hyperthermia such as genomic DNA repair, antioxidant response, protein synthesis, autophagy, and tissue growth [49]. These biological functions are associated with mechanisms that regulate the p53-signaling pathway and TORC1-signaling pathway [50]. The TORC1 complex integrates environmental and nutritional external signals within the cell [51] and influences growth, proliferation and metabolism [52]. TORC1 pathway is also responsible for cellular processes required for heat stress survival such as autophagy, cell cycle, mRNA stability and formation of ribosome components [53]. In a transcritpomic study, milking cows exposed to heat stress enhanced phosphorylation of mTORC1, which activated protein translation in mammary glands to maintain milk protein synthesis [54].

The gene *MDM4* (MDM4 Regulator of p53) enhances both stabilization and activation of p53-signal pathway. Under stress conditions, this gene is able to stimulate and antagonize p53 protein degradation because it has the ability to potentiate or reduce functionality of MDM2, the main ubiquitin ligase that regulates p53 [55]. However, the *MDM4* gene appears to play a critical role in regulation of the feedback loop between p53 and MDM2 [56]. A recent transcriptomic study from bovine granulose cells exposed to acute heat stress detected the *MDM4* gene as part of a complex signaling pathway, which affected granulose cells’ proliferation and caused a high apoptosis rate [57].

The Forkhead box O (FoxO) signaling pathway induces expression of multiple pro-apoptotic factors to promote inhibition of cell growth and/or apoptosis signaling. Transcriptional functions of FoxO are inhibited by the Akt pathway to favor cell survival, growth and proliferation [58]. Under heat stress conditions, upregulated expression of HSP72 induces phosphorylation of FoxO signaling [59]. In the current study, we found that the FoxO signaling pathway was also significantly enriched by DEGs. The genes in this pathway were *ATM* and *TGFβR1,* which were upregulated in HT ewes. Also were in this pathway the genes *IL10* and *IL6ST*, which were downregulated but showed low connectivity and limited gene interaction.

The transforming growth factor β (TGFβ) family regulates several cellular functions such as proliferation, survival, apoptosis, autophagy, dormancy and senescence. TGFβ signaling cascade starts with phosphorylation of TGFβ type I receptor (TGFβR1) and subsequent SMADs’ activation. Then, SMADs translocate into the nucleus to regulate expression of particular target genes [60]. Pro-apoptotic effects of TGFβ have been consistently reported, even though it is able to either induce or suppress apoptosis [61]. After exposure to chronic heat stress, the TGFβ pathway has been associated with alterations in cell architecture [62]. Under these conditions, activation of heat shock protein 70 (HSP70) appeared to reduce TGFβ signaling, making the cells more resistant to heat shock [63].

### 4.2. Immune Response

The immune system is activated in ewes exposed to moderate heat stress to favor adaptation to the environment and ensure physiological performance. However, when heat stress is severe or extreme, the immune system is weakened, which increases their susceptibility to diseases [13]. Under such conditions, genes and signaling pathways are activated to regulate animal response to heat stress.

The Retinoic acid-inducible I (RIG-I)-like receptor-signaling pathway was significantly enriched in the current study as a result of upregulated expression of *ISG15*, *IRF7* and *DHX58* genes in the HT group. This pathway mediates the transcriptional induction of type 1 interferon (Type-1 IFN), a potent cytokine that protect the organism against viral infections, but is also highly sensitive to heat stress. Similarly, interferon-stimulated genes (ISGs) are affected by heat stress [64]. After binding to its receptor, type-1 IFN activates the JAK-STAT pathway through phosphorylation of STAT1 and STAT2. Once activated, both signal transducers combine with interferon regulatory factors (IRF) to produce the interferon-stimulated gene factors (ISGF), which drive the expression of the ISGs [65]. 

The interferon-stimulated gene 15 (*ISG15*) plays a critical role in the innate immune response against microbial and viral infections through regulating cytokine release and suppressing IFN signaling. *ISG15* is able to bind to a wide range of target proteins through ISGylation, a ubiquitin-like post-translational modification encoded by *ISG15* [66]. Interestingly, the function of *ISG15* in non-viral innate immunity has been observed in response to external stimuli such as solar radiation, ischemia and DNA damage, which are consequences of heat stress exposure [67]. Furthermore, *ISG15* functions as a ubiquitin-like protein because it contains two ubiquitin-like domains with an amino acid sequence highly similar (50%) to ubiquitin [68,69]. The ubiquitin function is positively correlated with thermotolerance, as this function is activated during heat stress exposure, helping to remove damaged or unfolded proteins, avoiding its accumulation and conferring the cells’ ability for heat tolerance [70].

The IFN regulatory factor 7 (*IRF7*) is a master transcription factor for type-I IFN genes. Once phosphorylated, *IRF7* transcribes multiple type-I IFN genes by assembling into an IRF7 homodimer. Under stressed environments, the increased expression of the inducible Heat shock protein 70 (HSP70) appeared to display a negative regulation on the *IRF7* expression [71]. In mammalian cells exposed to metabolic and environmental stresses such as nutrient deprivation and ROS generation, a complex cellular response is induced and leads to phosphorylation of the eukaryotic factor 2a (eIF2a). This elevates the translation of activating transcription factor 4 (ATF4), a key component in the integrated stress responses that inhibit *IRF7* activation [72]. A reduced *IRF9* expression was reported in early pregnant cows subjected to heat stress [64].

Pregnant ewes exposed to sustained heat stress showed a reduction in embryo survival ability and pregnancy maintenance, leading to a decrease in lambing rates [73]. During mid- and late pregnancy, exposure to warm ambient temperatures impairs placental function, decreases fetal growth, reduces lamb birth weight and decreases lamb survival [74,75]. Interestingly, genes associated with thermotolerance in pregnant ewes have also been reported as markers for fertility, suggesting a close relationship between the genetic ability to tolerate heat stress and the reproductive performance. In this regard, the genes *PAM*, *STAT1* and *FBXO11* were reported as markers for heat stress tolerance in Columbia–Rambouillet pregnant ewes [9,76]. These three genes were later confirmed as markers associated with reproductive traits and physiological variables indicative of thermotolerance in Pelibuey pregnant ewes [10].

The DExH-Box Helicase 58 (*DHX58*) gene is involved in the negative regulation of type-I IFN production. Reduced *DHX58* expression has been reported in fish exposed to water at a warm temperature [77]. In addition, the *DHX58* gene was detected as a hub gene from DEGs in response to heat stress exposure in Ethiopian chickens [31]. In rabbits exposed to acute stress, a robust response in expression of type-I IFN was observed and attributed to the *DHX58* gene, which regulates IFIH1 and DDX58 signal to modulate the IFN response [78]. 

Finally, network analyses revealed the genes *ISG15, IRF7* and *DHX58* as largely connected and showing a high number of gene interactions, which highlighted the regulatory ability of these genes in ewes exposed to heat stress. In addition, co-expression networks confirmed the functional relationship among these three genes in white blood cells. Together, these results suggested that a genetically mediated non-viral immune response against environmental sense stressors that involves apoptosis appears to play an important role in tolerance to heat stress in sheep.

## 5. Conclusions

Physiological responses to heat stress are complex processes underpinned by genetic regulation. In this study, we compared transcriptomic changes in blood from heat-stressed ewes that were classified as heat tolerant and non-heat tolerant. Our work revealed differentially expressed genes and functional pathways associated with apoptosis, cell signaling and immune response to external stressors. Co-expression networks confirmed immune-related genes within WBCs, which were consistent with hematological findings. These results indicated that a non-viral immune mechanism appears to underlie the improved genetic ability to tolerate heat stress. This provides evidence helpful to elucidate the molecular mechanisms involved in thermotolerance in ewes exposed to a heat-stressed environment. However, further validation studies in independent populations are required to confirm our candidate genes as molecular markers for heat stress tolerance in ewes.

## Figures and Tables

**Figure 1 genes-14-01590-f001:**
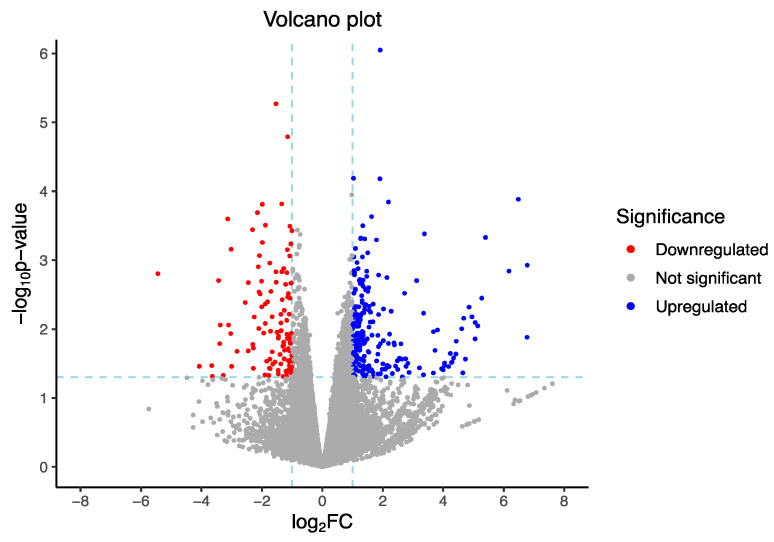
Volcano plot showing differentially expressed genes between HT and NHT ewes.

**Figure 2 genes-14-01590-f002:**
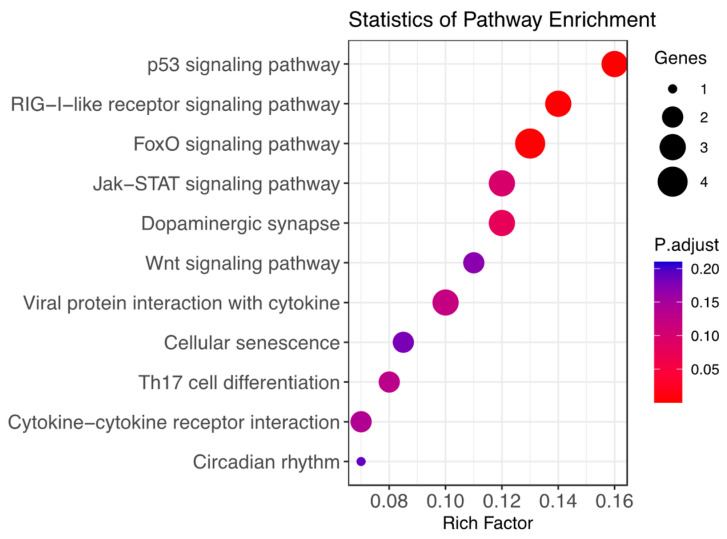
Functional enrichment analysis of DEGs between heat tolerant (HT) and non-heat tolerant ewe’s groups: KEGG pathway analysis.

**Figure 3 genes-14-01590-f003:**
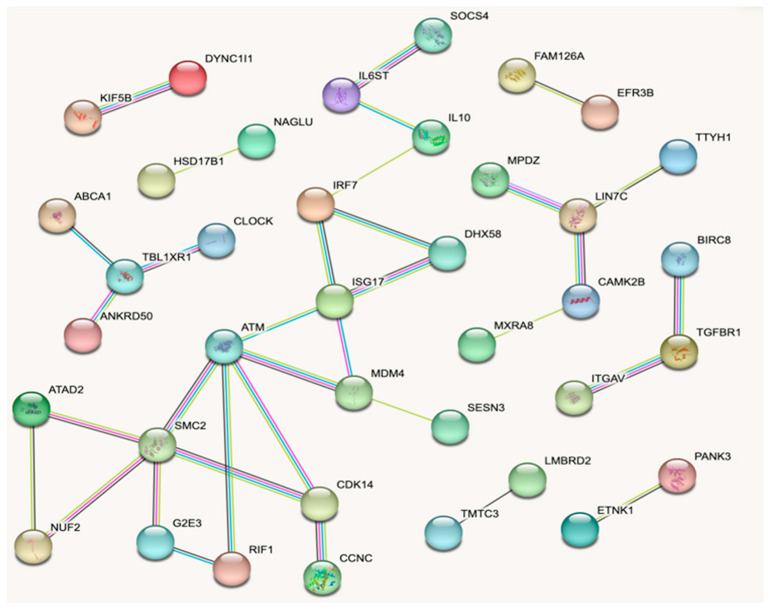
Gene network from STRING database displaying the connections between the main DEGs (i.e., more than one connection) detected between HT and NHT ewes exposed to heat stress.

**Figure 4 genes-14-01590-f004:**
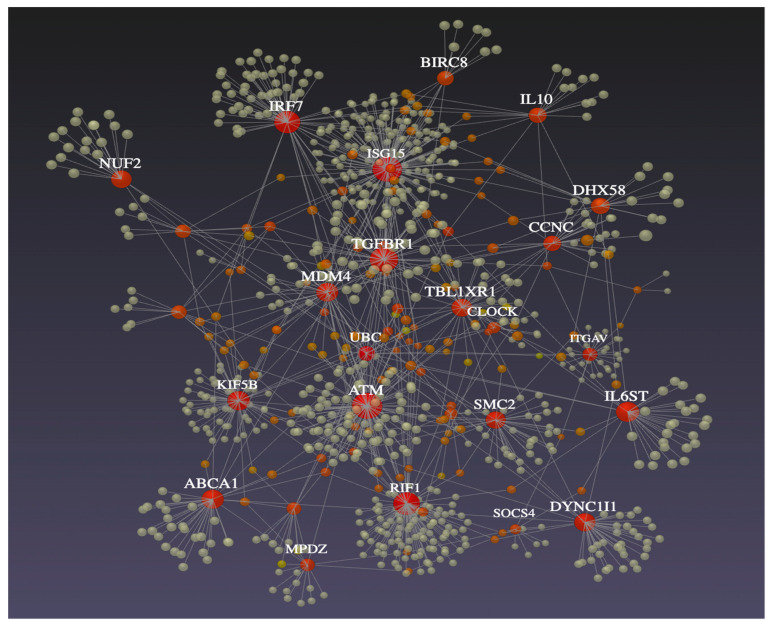
IMEx Interactome gene network from NetworkAnalyst displaying the interactions between the main DEGs (i.e., more than 30 interactions) detected between HT and NHT ewes exposed to heat stress.

**Figure 5 genes-14-01590-f005:**
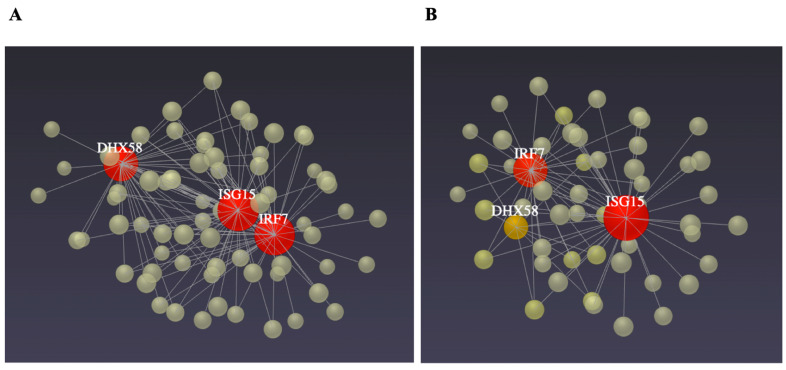
Co-expression gene networks from NetworkAnalyst displaying DEGs within white blood cells from heat-stressed ewes. (**A**) Lymphocytes and (**B**) Monocytes.

**Figure 6 genes-14-01590-f006:**
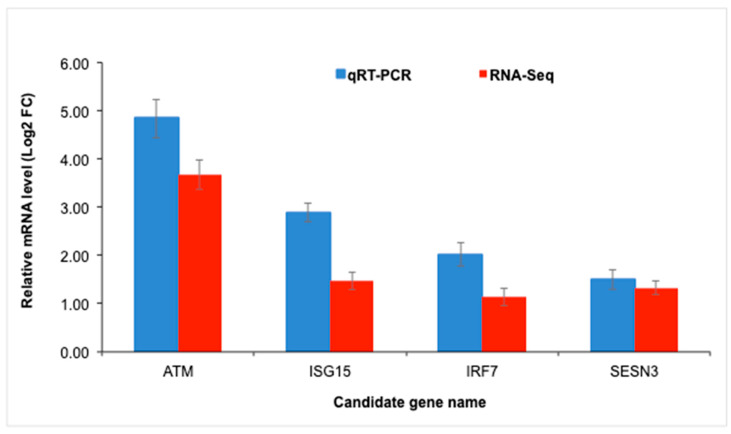
Quantitative real-time PCR (qRT-PCR) validation of four differentially expressed genes (DEGs). Blue bar values are the means for qRT-PCR (*n* = 14) and red bar values are the means for RNA-Seq (*n* = 16). A significant correlation (R = 0.95; *p* < 0.05) was observed between the fold changes determined by qRT-PCR and RNA-Seq for the 4 genes examined in the study.

**Table 1 genes-14-01590-t001:** Regression analysis performed to classify heat-stressed ewes as heat tolerant (HT) and non-heat tolerant (NHT), according to the regression coefficient (β1).

Heat Tolerant (HT) Ewes				Non-Heat Tolerant (NHT) Ewes			
ID ^1^	β_1_ ^2^	R2 ^3^	*p*-value ^4^	ID ^1^	β_1_ ^2^	R2 ^3^	*p*-Value ^4^
Y18	0.8902	0.5922	<0.0001	5	−0.8194	0.4095	0.0216
A44	0.7759	0.5012	0.0002	844	−0.5273	0.5476	0.0002
R40	0.4918	0.3595	0.0485	831	−0.4058	0.5569	<0.0001
845	0.3710	0.5712	0.0005	1	−0.3939	0.4340	0.0065
810	0.3606	0.4281	0.0092	826	−0.3595	0.5227	0.0004
815	0.3160	0.4931	0.0013	816	−0.3136	0.4816	0.0018
858	0.2291	0.4911	0.0002	822	−0.2857	0.5670	0.0003
817	0.1522	0.4576	0.0076	819	−0.1066	0.4133	0.0314

^1^ Sheep ID (ear tag); ^2^ Regression coefficient; ^3^ Coefficient of determination; ^4^ Statistical significance for the regression analysis.

**Table 2 genes-14-01590-t002:** Comparison of the hematological values between heat tolerant (HT) and non-heat tolerant (NHT) ewes exposed to heat stress.

		Heat Stressed Groups		
Blood Trait ^1^	Normal Range ^2^	HT ^3^	NHT ^4^	*p*-Value ^5^
WBC	4.0–12.0 × 10^3^/μL	6.04 ^a^	7.81 ^b^	0.024
RBC	9.0–15.0 × 10^6^/μL	11.16 ^a^	9.65 ^b^	0.021
HGB	9.0–18.0 g/dL	11.18	10.64	0.632
HCT	27.0–45.0%	35.38	32.73	0.131
MCV	28–40 fL	33.10	32.30	0.489
MCH	8.0–12.0 pg	10.49	10.65	0.642
MCHC	31–34 g/dL	31.68	32.74	0.391
Platelets	240–700 × 10^3^/μL	511.75	504.63	0.920
Neutrophils	700–6000/μL	2890.75	2742.25	0.819
Lymphocytes	2000–10,000/μL	2615.75 ^a^	4388.35 ^b^	<0.001
Monocytes	100–300/μL	134.21 ^a^	203.63 ^b^	0.026
Eosinophil	0–1000/μL	200.38	278.88	0.099
Basophils	0/μL	22.88	3.75	0.187

^1^ Blood components (WBC = White blood cells; RBC = Red blood cells; HGB = Hemoglobin; HCT = Hematocrit; MCV = Mean cell volume; MCH = Mean corpuscular hemoglobin; MCHC = Mean corpuscular hemoglobin concentration); ^2^ Normal range for hematological values in sheep; ^3^ Hematological average values for HT ewes; ^4^ Hematological average values for NHT ewes; ^a,b^ Indicate statistical difference between HT and NHT hematological average values; ^5^ *p*-value from the Student’s *t*-test for the comparisons between HT and NHT animals.

**Table 3 genes-14-01590-t003:** Significantly enriched gene ontology terms associated with DEGs in heat tolerant (HT) and non-heat tolerant (NHT) ewes exposed to heat stress.

Gene Ontology (GO) Term ^1^	GO Group ^2^	No. DEGs ^3^	*p*-Value ^4^
Regulation of TORC1 signaling	BP	15	5.3 × 10^−4^
Regulation of cell communication	BP	15	6.4 × 10^−4^
Type I interferon production	BP	11	6.6 × 10^−4^
Regulation of signal transduction	BP	14	6.9 × 10^−4^
Regulation of signaling	BP	12	1.7 × 10^−3^
Regulation of immune system process	BP	10	2.9 × 10^−3^
Lipoprotein metabolic process	BP	11	3.7 × 10^−3^
Regulation of cytokine production	BP	10	4.2 × 10^−3^
Cellular metabolic process	BP	10	6.4 × 10^−3^
Regulation of response to stress	BP	9	1.7 × 10^−2^
Bicellular tight junction	CC	3	1.8 × 10^−2^
External side of plasma membrane	CC	3	4.6 × 10^−2^
ATP binding	MF	10	4.1 × 10^−4^
Cytokine binding	MF	10	1.5 × 10^−3^

^1^ Gene ontology annotation of the DEGs; ^2^ Gene ontology groups (BP = Biological process; CC = Cellular component; MF = Molecular function); ^3^ Number of DEGs; ^4^ Statistical significance.

**Table 4 genes-14-01590-t004:** KEGG pathway enrichment analysis based on the differentially expressed genes (DEGs) between heat tolerant (HT) and non-heat tolerant (NHT) ewes.

KEGG Pathway ^1^	Main Genes ^2^	*p*-Value ^3^	Corrected *p*-Value ^4^
p53 signaling pathway	ATM, SESN3, MDM4	2.7 × 10^−4^	1.8 × 10^−3^
RIG-I-like receptor signaling pathway	ISG15, IRF7, DXH58	7.4 × 10^−4^	2.5 × 10^−3^
FoxO signaling pathway	TGFBR1, ATM, IL10, IL6ST	1.1 × 10^−3^	4.3 × 10^−3^
Dopaminergic synapse	KIF5B, CAMK2B, CLOCK	2.8 × 10^−3^	7.6 × 10^−2^
Jak-STAT signaling pathway	IL10, SOCS4, IL6ST	5.1 × 10^−3^	9.7 × 10^−2^
Viral protein interaction with cytokine	IL10, TGFBR1, IL6ST	8.7 × 10^−3^	1.2 × 10^−1^
Th17 cell differentiation	TGFBR1, IL6ST	1.3 × 10^−2^	1.3 × 10^−1^
Cytokine–cytokine receptor interaction	IL10, TGFBR1, IL6ST	1.6 × 10^−2^	1.4 × 10^−1^
Wnt signaling pathway	TBL1XR1, CAMK2B	2.5 × 10^−2^	1.7 × 10^−1^
Cellular senescence	TGFBR1, ATM	2.6 × 10^−2^	1.8 × 10^−1^
Circadian rhythm	CLOCK1	3.4 × 10^−2^	1.9 × 10^−1^

^1^ KEGG pathways involved in heat stress tolerance; ^2^ Main genes associated with KEGG pathways; ^3^ Statistical significance; ^4^ Statistical significance after FDR correction.

**Table 5 genes-14-01590-t005:** Top 10 genes with the highest regulatory potential according to their connectivity (STRING) and the number of interactions (NetworkAnalyst).

Gene Symbol	No. Connec. ^1^	Gene Symbol	No. Interac. ^2^
ATM	5	ISG15	188
SMC2	5	RIF1	120
ISG15	4	ATM	117
IRF7	3	KIF5B	73
MDM4	3	IRF7	66
CDK14	3	TGFBR1	65
TBL1XR1	3	MDM4	48
LIN7C	3	DYNCIII	41
DHX58	2	ABCA1	40
TGFBR1	2	DHX58	37

^1^ Number of gene connections according to STRING analysis; ^2^ Number of interactions according to NetworkAnalyst data.

## Data Availability

The data presented in this study are available on reasonable request from the corresponding author.

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
