# Peer review of "Blood Transcriptomic Analyses Reveal Functional Pathways Associated with Thermotolerance in Pregnant Ewes Exposed to Environmental Heat Stress"

_genes, 2023, doi:10.3390/genes14081590_

Round 1

Reviewer 1 Report

The manuscript is interesting. But I think it is not deep enough, and there is many problems. Some problems are listed below:

1. One-way ANOVA is not suitable for comparing differences between two groups.

2. Line60, differential expressed genes should be differentially expressed genes. Here, abbreviation “DEG” has been used, so I think it is better to used DEG thereafter, such as in lines 175 and 170.

3. Section 2.4, “RNA isolation and sequencing”, please describe clearly how to select animals and perform RNA-seq, that is, at each time point, is there repeats, and how to group the animals.

4. Figure 2, I think rich factor is better than count in X-axis

5. Table 3, the title does not seem to tally with the contents in the table

6. Lines 175-177, the relationship between p-value and FDR was not depicted clearly, when was p-value or FDR was used?

7. Provide the website address of STRING database and other network resources used.

8. Line 204, “design synthetic oligonucleotide primers” should be “design oligonucleotide primers”

9. Line 211, “geometric gene”, what is the meaning?

10. Figure 1, there is no red circle.

...

Author Response

We thank the Reviewers for all comments and suggestions that helped to improve the quality of the manuscript. We have answered to each individual comments and suggestions made by the Reviewers (below). All revisions made to the manuscript were highlighted in yellow.

  1. -way ANOVA is not suitable for comparing differences between two groups.

Response: Thank you for your suggestion, we used the Student’s T-Test instead of One-way ANOVA to compare differences in hematological values between the two groups. P-values for each comparison were updated in Table 2.

  1. Line 60, differential expressed genes should be differentially expressed genes.

Response: We have made this correction.

Here, abbreviation “DEG” has been used, so I think it is better to used DEG thereafter, such as in lines 175 and 170.

Response: We have made this correction through the manuscript.

  1. Section 2.4, “RNA isolation and sequencing”, please describe clearly how to select animals and perform RNA-seq, that is, at each time point, is there repeats, and how to group the animals.

Response: We have made these corrections in section 2.3 (formerly section 2.4).

  1. Figure 2, I think rich factor is better than count in X-axis.

Response: We have replaced count by rich factor in X-axis from Figure 2 (formerly Figure 2B). We have removed Figure 2A because this information is already included in Table 3.

  1. Table 3, the title does not seem to tally with the contents in the table.

Response: We have corrected the title of Table 3.

  1. Lines 175-177, the relationship between p-value and FDR was not depicted clearly, when was p-value or FDR was used?

Response: We have clarified that FDR was performed to adjust P-values of genes differentially expressed (DEGs), and a threshold of adjusted P-value < 0.05 was used to identify significant DEGs.

  1. Provide the website address of STRING database and other network resources used.

Response: We have added in section 2.6 the website address of STRING database and all other network resources used.

  1. Line 204, “design synthetic oligonucleotide primers” should be “design oligonucleotide primers”.

Response: We have made this correction.

  1. Line 211, “geometric gene”, what is the meaning?

Response: We have replaced “geometric gene” by “geometric mean”. I apologize because it was a typo error.

  1. Figure 1, there is no red circle.

Response: We have replaced the Volcano plot with another one that is more consistent with the data shown in the text of the manuscript (i.e., Volcano plot showing down- and up-regulated DEGs).

Reviewer 2 Report

The authors have taken a good effort to conduct this study and draft the manuscript. This is an interesting study however I have a few minor comments/suggestions as indicated below

1.      From what I could understand, the hematological variables and mRNA sequencing were done on different set of animals (in line 132, the authors have stated, ‘Another subset of 16 pregnant ewes exposed to heat stress…’), can the authors explain the reason for doing so?

2.      What was the total number of animals used in the current study, before categorizing them into heat tolerant and non-heat tolerant? Was is 16/17 or 32? For the hematological variables, the total animal count comes to 17 while the same for mRNA sequencing was stated to be 16. Additionally the authors have also mentioned that ‘another subset of pregnant ewes’ were used for the RNA sequencing study (line 132). Hence it’s a confusing to understand the total animal count. Kindly look into this

3.      Does the linear regression analysis used to assess the thermo-tolerance indicator also take into consideration the pregnancy/gestation stage of the animal as a factor? Though all the animals were in the similar stage of gestation, pregnant and non-pregnant animals exhibit varied intensity of response to heat stress

4.      Line 119: the total number of animals for HT group is indicated as 9, was this a typing error? The number of animals indicated in table 1 was 8 + 8 therefore total 16. Kindly confirm this

5.      Table 4: kindly indicate all gene names in italics font, follow the same throughout the manuscript

6.      The study was done on pregnant ewes however neither the results point towards any negative impact on fertility or reproduction related traits nor have the authors discussed the results in this line. Hence I suggest to include this aspect in the discussion too.

Author Response

We thank the Reviewers for all comments and suggestions that helped to improve the quality of the manuscript. We have answered to each individual comments and suggestions made by the Reviewers (below). All revisions made to the manuscript were highlighted in yellow.

  1. From what I could understand, the hematological variables and mRNA sequencing were done on different set of animals (in line 132, the authors have stated, ‘Another subset of 16 pregnant ewes exposed to heat stress…’), can the authors explain the reason for doing so?

Response: I kindly confirm you that, yes, we used different set of animals to perform mRNA sequencing and hematological studies.

RNA sequencing was performed in blood samples collected from pregnant ewes on days -1, 2, 4, 7, 14 and 28 of heat stress exposure. We detected the highest differential gene expression between heat tolerant (HT) and non-heat tolerant (NHT) sheep groups at day 7 after heat exposure begins, which suggested that significances changes in blood parameters may be occurring by day 7. In order to confirm that and for obtaining more information useful to explain the differential gene expression, we decided to collect blood samples for hematological studies in another set of pregnant ewes at day 7 after starting heat stress exposure.

In order to be consistent with the methodology used in the current research, we have reordered “Hematological parameters” section to the end of Materials and Methods.

  1. What was the total number of animals used in the current study, before categorizing them into heat tolerant and non-heat tolerant? Was is 16/17 or 32? For the hematological variables, the total animal count comes to 17 while the same for mRNA sequencing was stated to be 16. Additionally the authors have also mentioned that ‘another subset of pregnant ewes’ were used for the RNA sequencing study (line 132). Hence it’s a confusing to understand the total animal count. Kindly look into this.

Response: Thank you for your comment and I apologize for the confusion. We have reordered “Hematological parameters” section to the end of Materials and Methods, which will help to clarify the total animal count. In summary, we used two sets of pregnant ewes according to the performed studies: 1) The mRNA sequencing study that included 16 ewes (8 HT and 8 NHT) as mentioned in the section “2.3. RNA isolation and sequencing”, and 2) The hematological study that included 17 ewes (9 HT and 8 NHT) as mentioned in the section “2.8. Hematological parameters”.

  1. Does the linear regression analysis used to assess the thermo-tolerance indicator also take into consideration the pregnancy/gestation stage of the animal as a factor? Though all the animals were in the similar stage of gestation, pregnant and non-pregnant animals exhibit varied intensity of response to heat stress.

Response: Thank you for your comment. Gestation age and initial body weight were included in the regression model, but these variables did not explain a significant variation in the response trait.

We used the pregnant model because the increased nutrient demand and high metabolic rate make pregnant ewes more sensitive to heat stress. Also, physiological studies that were performed in our lab over the last 10 years have detected variation in the degree of the pregnant ewes’ responses to heat stress. Recent reports from our group have revealed pathways and genes associated with thermotolerance suggesting a genetic component underlying heat stress tolerance in pregnant ewes. Collectively these data provided supporting evidence to propose the pregnant ewe as a valuable model to study genetic basis associated with tolerance to heat stress in livestock.

  1. Line 119: the total number of animals for HT group is indicated as 9, was this a typing error? The number of animals indicated in table 1 was 8 + 8 therefore total 16. Kindly confirm this.

Response: It was not a typing error. To clarify the text, line 119 is talking about the set of ewes used for hematological variables (i.e., 9 HT and 8 NHT), whereas Table 1 refers to ewes used for the mRNA sequencing study (i.e., 8 HT and 8 NHT).

  1. Table 4: kindly indicate all gene names in italics font, follow the same throughout the manuscript

Response: We have made this correction throughout the manuscript.

  1. The study was done on pregnant ewes however neither the results point towards any negative impact on fertility or reproduction related traits nor have the authors discussed the results in this line. Hence I suggest to include this aspect in the discussion too.

Response: We have added, near the end of the discussion, some information regarding to the negative impact of heat stress on fertility in pregnant ewes, as well as other information related to genetic basis associated with these variables.

In the current study, ewes were bred and maintained in thermoneutral conditions until exposed to heat stress conditions at ~39 days of gestation.

Round 2

Reviewer 1 Report

There are some minor spelling errors such as RNAseq (should be RNA-seq), etc.

Discussion should be shorten.